# Mismatch repair-signature mutations activate gene enhancers across human colorectal cancer epigenomes

Stevephen Hung[1], Alina Saiakhova[1], Zachary J Faber[1], Cynthia F Bartels[1], Devin Neu[1], Ian Bayles[1], Evelyn Ojo[2], Ellen S Hong[1], W Dean Pontius[3], Andrew R Morton[1], Ruifu Liu[2], Matthew F Kalady[3,4,5], David N Wald[2,6], Sanford Markowitz[1,6,7], Peter C Scacheri[1,6]*

[1]Department of Genetics and Genome Sciences, Case Western Reserve University, Cleveland, United States; [2]Department of Pathology, Case Western Reserve University, Cleveland, United States; [3]Department of Molecular Medicine, Lerner Research Institute, Cleveland Clinic, Cleveland, United States; [4]Department of Cancer Biology, Lerner Research Institute, Cleveland Clinic, Cleveland, United States; [5]Department of Colorectal Surgery, Digestive Disease and Surgery Institute, Cleveland Clinic, Cleveland, United States; [6]Case Comprehensive Cancer Center, Case Western Reserve University, Cleveland, United States; [7]Department of Medicine, Case Western Reserve University, Cleveland, United States

*For correspondence:
pxs183@case.edu

Competing interests: The authors declare that no competing interests exist.

**Abstract** Commonly-mutated genes have been found for many cancers, but less is known about mutations in cis-regulatory elements. We leverage gains in tumor-specific enhancer activity, coupled with allele-biased mutation detection from H3K27ac ChIP-seq data, to pinpoint potential enhancer-activating mutations in colorectal cancer (CRC). Analysis of a genetically-diverse cohort of CRC specimens revealed that microsatellite instable (MSI) samples have a high indel rate within active enhancers. Enhancers with indels show evidence of positive selection, increased target gene expression, and a subset is highly recurrent. The indels affect short homopolymer tracts of A/T and increase affinity for FOX transcription factors. We further demonstrate that signature mismatch-repair (MMR) mutations activate enhancers using a xenograft tumor metastasis model, where mutations are induced naturally via CRISPR/Cas9 inactivation of *MLH1* prior to tumor cell injection. Our results suggest that MMR signature mutations activate enhancers in CRC tumor epigenomes to provide a selective advantage.
DOI: https://doi.org/10.7554/eLife.40760.001

## Introduction

In the past decade, tumor sequencing efforts by TCGA, ICGC, and others have identified the most frequently mutated driver genes in most common forms of cancer (*Lawrence et al., 2014*; *Kandoth et al., 2013*). Outside protein-coding genes, mutations in non-coding regions that disrupt enhancer-gene control are emerging as a major mechanism of cancer development. Examples include large chromosomal rearrangements that hijack enhancers to oncogenes, such as the Burkitt lymphoma translocation that repositions IgH enhancers upstream of *MYC* (*Battey et al., 1983*). Copy number alterations can amplify enhancer sequences near oncogenes. Deletions can remove boundaries between enhancers and proto-oncogenes, and inversions can flip enhancers to proto-oncogenes (*Zhang et al., 2016*; *Beroukhim et al., 2016*; *Hnisz et al., 2016*). Besides large structural variants that rewire gene-enhancer interactions, small-scale mutations that lie *within* regulatory elements and alter their activity can occur. The first discovered were recurrent point mutations in the

*TERT* promoter in melanoma and other cancers (*Huang et al., 2013*). Other examples include an indel in T-ALL that creates a super-enhancer that drives overexpression of the *TAL1* oncogene (*Mansour et al., 2014*), and recurrent enhancer substitutions and indels that affect the expression of *PAX5* in CLL (*Puente et al., 2015*). The discovery of these driver events has motivated searches for additional enhancer mutations in other common cancers, but so far their prevalence and relevance to the cancer phenotype remain largely undetermined.

The identification of functional enhancer mutations is challenging due to several confounding factors. First, mutation rates vary considerably between different tumor types and even among tumors of the same subtype. Second, tumor epigenomes are heterogeneous and mutation rates are profoundly influenced by chromatin states, with euchromatic, early-replicating regions showing a low mutation rate relative to heterochromatic, late-replicating regions (*Schuster-Böckler and Lehner, 2012*; *Polak et al., 2015*). Given the variation, the conventional approach of overlaying mutations detected through tumor sequencing with a 'reference' epigenome is suboptimal. Strategies that facilitate simultaneous capture of both sequence content and regulatory activity are more suitable. Third and perhaps most importantly, for many cancers the cell type of origin is unknown or unavailable for epigenomic studies. The lack of the normal comparator makes it difficult to assess whether a putative mutation influenced the activity of the regulatory element relative to the normal cell from which the tumor was derived.

Through ChIP-seq analysis of enhancer histone marks (H3K4me1 and H3K27ac), we previously compared the enhancer epigenomes of a genetically-diverse cohort of human CRC models to normal colonic crypts, the cell type of origin for CRC. We identified Variant Enhancer Loci (VELs) as sites that differed in the levels of H3K4me1 and H3K27ac between normal crypts and CRC (*Akhtar-Zaidi et al., 2012*; *Cohen et al., 2017*). Here, we pinpoint functional enhancer mutations in VELs directly from H3K27ac ChIP-seq data, using the logic that a DNA variant in an enhancer with higher H3K27ac levels in CRC than normal may have contributed to the activation of that 'gained' enhancer. Our analysis shows that CRC samples with underlying deficiencies in mismatch repair harbor an exceptionally high indel rate in gained enhancers compared to their already high background mutation rate. We provide evidence that these non-coding mutations, previously presumed to be passengers, are functional.

## Results

### Identification of putative enhancer activating indels

We looked for candidate mutations that augment enhancer activity by identifying somatic mutations in regions with elevated levels of H3K27ac in CRC relative to normal colon (*Figure 1a*). A key step in the analysis is identifying instances of allele bias, where H3K27ac ChIP-seq read depth is higher on the allele containing the mutation than on the reference allele. We further eliminate mutations that are not predictive of gained H3K27ac enrichment (i.e., the mutation occurs in a cell line with the gained enhancer, but not in other cell lines with that same enhancer), as these are likely passenger events. We focused on indels because they have previously been shown to stimulate enhancer activation through *de novo* creation of transcription factor binding sites (*Mansour et al., 2014*). Using H3K27ac ChIP-seq data from 24 cell lines derived from all clinical stages of CRC, we detected a total of 355 candidate enhancer-activating indels (example shown in *Figure 1b*). H3K27ac and H3K4me1 levels were highly enriched at the candidate enhancer-activating indels relative to these sites in normal colon (*Figure 1—figure supplement 1a–b*). Eighty five percent were located distal (>5 kb) to transcription start sites (*Figure 1c*), 76% overlapped H3K4me1 peaks, and 73% were located <1 kb from a DNase I hypersensitive site. Nineteen out of 20 indels detected through our analysis pipeline were validated through Sanger sequencing, indicating high specificity of our method (*Figure 1—figure supplement 2a–b*).

An analysis of the distribution of enhancer indel mutations across the full CRC cohort revealed two main classes, stratified by microsatellite stability (*Figure 1d*, *Figure 1—source data 1*). Microsatellite stable (MSS) samples harbored a lower enhancer mutation rate than MSI samples, which are deficient in mismatch repair (MMR) genes: *MLH1*, *MSH2, and PMS2*. The enhancer indels in MSI samples were predominantly short (1–2 bp) contractions of homopolymer runs of T's and A's (*Figure 1e*), the classic mutational signature found in coding regions of MSI tumors (*Ionov et al.,*

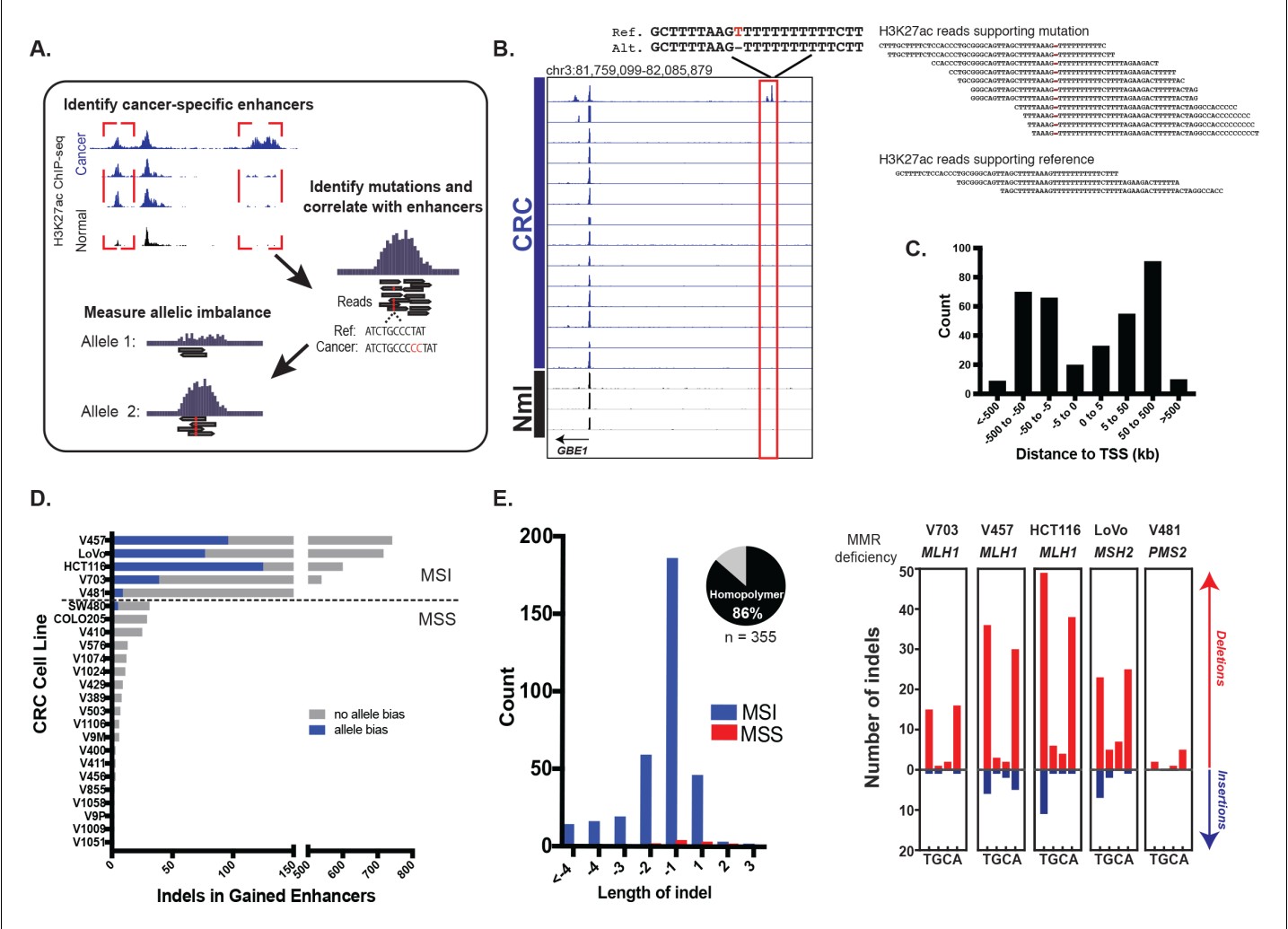

**Figure 1.** Identification of functional enhancer mutations from H3K27ac ChIP-seq data. (**A**) Overview of approach. (**B**) (left) Genome Browser snapshot of a putative enhancer-activating indel detected at a CRC-specific H3K27ac peak. Y-axis scales are all 0 to 25. (right) H3K27ac ChIP-seq reads covering indel locus. (**C**) Distribution of CRC-specific H3K27ac peaks containing indels relative to transcription start sites. (**D**) Number of gained enhancer-associated indels with and without allele bias detected in each CRC cell line. Enhancer mutations are provided in *Figure 1—source data 1*. (**E**) (left) Distribution of indel lengths detected in gained enhancers in MSI and MSS CRC samples. Pie chart shows the fraction of indels detected in homopolymers. (right) Number of insertion (blue) and deletion (red) mutations in homopolymers of T, G, C, or A in MSI cell lines.
DOI: https://doi.org/10.7554/eLife.40760.002

The following source data and figure supplements are available for figure 1:

**Source data 1.** Enhancer indel mutations detected from H3K27ac ChIP-seq data in cell lines and primary tumors.
DOI: https://doi.org/10.7554/eLife.40760.005
**Figure supplement 1.** Enhancer histone profiles at indel gained enhancers.
DOI: https://doi.org/10.7554/eLife.40760.003
**Figure supplement 2.** Validation of enhancer indels.
DOI: https://doi.org/10.7554/eLife.40760.004

*1993*; *Kim et al., 2013*). As expected, *MLH1* and *MSH2*-deficient cell lines also showed higher indel rates than *PMS2*-deficient cells (*Baross-Francis et al., 2001*; *Hegan et al., 2006*). To further test the relevance of these findings, we analyzed H3K27ac ChIP-seq data from four primary tumors (*Cohen et al., 2017*) of unknown microsatellite status. One of the four samples showed an elevated enhancer mutation rate and the signature MSI mutation of poly A/T homopolymers (*Figure 1—figure supplement 2c*). This was subsequently identified as the only MSI sample in the group, obtained from a Lynch syndrome patient. The results indicate that enhancer mutations are prevalent in MSI-

forms of CRC, and they show the same MMR signature as coding regions. The presence of enhancer mutations in both cell lines and primary tumors rules out an in vitro-specific mechanism of enhancer activation.

## MSI enhancer mutations show evidence of positive selection

Similar to previous studies correlating point mutation rates with active histones (*Makova and Hardison, 2015*; *Kim et al., 2013*), indel mutation rate and H3K27ac levels were anti-correlated in MSS CRC (Pearson r 95% CI [−0.63,−0.59]). In contrast, there was no overall correlation between indel rate and H3K27ac levels in the MSI samples (Pearson r 95% CI [−0.020, 0.031]) (*Figure 2a*, *Figure 2—figure supplement 1a–b*, *Figure 2—source data 1*), with both H3K27ac-enriched and depleted regions showing a mutation rate 68 times higher than in MSS CRC. This result is consistent

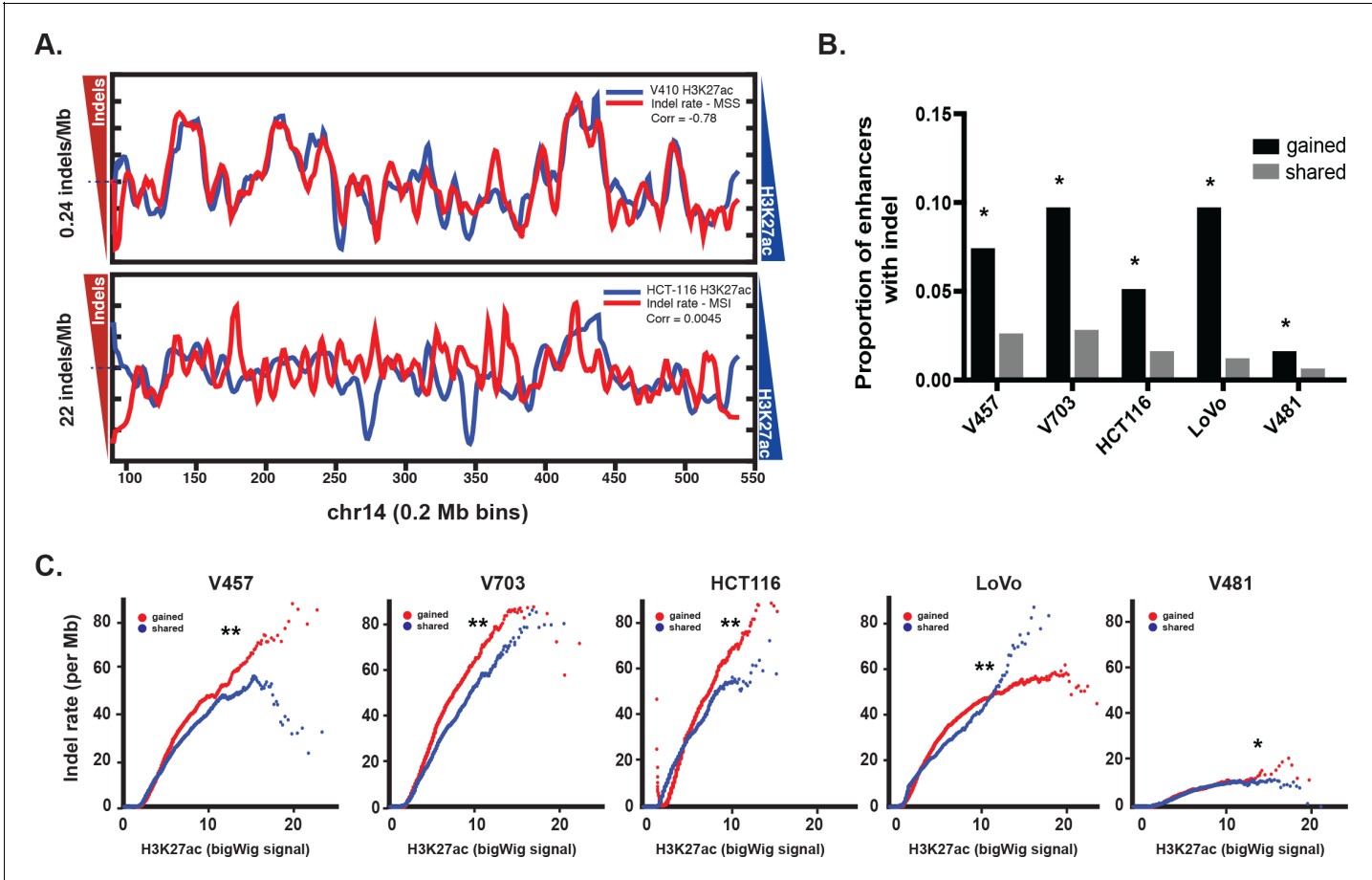

**Figure 2.** Enrichment of MSI indels in gained enhancers. (**A**) MSS (top) and MSI (bottom) mutations (red) are overlaid on H3K27ac profiles (blue) of representative MSS (V410) and MSI (HCT-116) lines in 0.2 Mb bins on chromosome 14. Median mutation rates are shown next to left Y axis. The H3K27ac signal is inverted to better show anti-correlation. Pearson correlation values are shown. (**B**) Proportion of gained enhancers (black) and enhancers shared with normal crypt (grey) with at least one allele-biased indel. *p<1e-10, Z-test for two proportions. (**C**) Rate of allele-biased indels in gained (red) and shared (blue) enhancers, as a function of H3K27ac signal. **, p<1e-10; *p<0.05, t-test (*Figure 2—source data 1*).
DOI: https://doi.org/10.7554/eLife.40760.006

The following source data and figure supplement are available for figure 2:

**Source data 1.** H3K27ac signal and mutation rate for MSS and MSI samples, in 0.5 Mb genomic bins.
DOI: https://doi.org/10.7554/eLife.40760.008

**Source data 2.** Indel rate and H3K27ac signal for gained enhancers and enhancers shared with crypt, for 5 MSI cell lines.
DOI: https://doi.org/10.7554/eLife.40760.009

**Figure supplement 1.** Genome-wide correlations of H3K27ac signal and mutation rate.
DOI: https://doi.org/10.7554/eLife.40760.007

with previous studies indicating that open regions of chromatin are no longer protected upon loss of MMR (*Supek and Lehner, 2015*). This raises the question whether gained enhancers in MSI CRC are a target for mutation simply because they lie in open chromatin, or if the mutations are truly functional. We reasoned that if the indels activate enhancers, then gained enhancers should show a higher proportion of allele-biased indels compared to enhancers that are already open, i.e, enhancers shared between CRC and normal crypt. In all 5 MSI samples, the proportion of allele-biased gained enhancers was higher than that of shared enhancers (*Figure 2b*). Even after controlling for enhancer length and H3K27ac signal intensity, gained enhancers were more likely to contain allele-biased indels than shared enhancers (*Figure 2c*, *Figure 2—source data 2*), indicating positive selection of the gained enhancers with indels.

## MSI enhancer mutations increase expression of their predicted gene targets

We next tested the effect of the indels in gained enhancers on target gene expression. To isolate the effect of the indel from that of the gained enhancer, we retrieved genes associated with gained enhancer indels, and compared their expression in cell lines with both the indel and the gained enhancer to that of cell lines with only the gained enhancer. Genes regulated by gained enhancers containing indels were more highly expressed than those same genes regulated by gained enhancers containing the wildtype sequence (*Figure 3a*, *Figure 3—source data 1*). We further verified that this expression difference was unlikely to be due to differences in the levels of H3K27ac between the two test groups (*Figure 3—figure supplement 1a*, *Figure 3—source data 2*). Gene Ontology terms significantly enriched among indel-associated genes included tissue development and embryogenesis, regulation of biosynthesis, and cell proliferation (*Figure 3b*). Collectively, the results suggest that the MSI enhancer indels lead to increased expression of target genes, several of which function in cell growth and early development and may therefore enhance fitness or contribute to changes in cell state.

## MSI enhancer mutations are recurrent

To test if any of the enhancer mutations were recurrent, we analyzed whole genome sequence data from 47 primary CRC samples (TCGA), 3 CRC cell lines (in house), and their matched normal DNAs. In total, we detected 1,111,539 somatic substitutions and 374,859 indels in the tumor enhancerome. Mutation rates in the enhancerome were similar to those reported for the exome (*Lawrence et al., 2014*), with a clear separation of hypermutator and non-hypermutator samples (*Figure 3—figure supplement 1b*). Of the 16 hypermutated samples, 10 had MMR-mutational signatures and were designated MSI, and six had POLE-mutational signatures (*Alexandrov et al., 2013*). Thirteen percent (46/355) of enhancer indels detected in the CRC cell lines were found to be recurrent in at least one of the 50 WGS samples. Fifteen indels were significantly recurrent in two or more primary CRC samples (*Figure 3c*, *Figure 3—source data 1*). The recurrent indels were found predominantly in MSI samples, and also occasionally in MSS samples with elevated mutation rates, suggesting that enhancer activation by indels occurs whenever a sufficiently high mutation rate is reached. Furthermore, consistent with their positive selection, recurrent indels occurred more frequently in gained enhancers than in enhancers shared between tumor and normal crypts (*Figure 3—figure supplement 1c*). Genes associated with the most recurrent indels (3–4 of 50 samples) include *UBE2V2*, a ubiquitin-conjugating enzyme linked to chemoresistance (*Santarpia et al., 2013*), *USP8*, a deubiquitinase linked to EGFR signaling (*Kim et al., 2017*), and *COPB2*, a vesicle coating protein linked to regulation of cell proliferation and apoptosis (*Mi et al., 2016*) (*Figure 3c*). The top-enriched GO terms for recurrent enhancer indel genes reflect known cancer hallmarks, including cell proliferation, signal transduction, and cell communication (*Figure 3d*).

We next leveraged data from gene-knockdown studies from the DepMap project to test if the recurrent enhancer genes affect cell viability. As a set, these genes were not significantly enriched for dependency genes (p=0.15). However, among the set, genes associated with the GO term 'cell proliferation' showed significantly lower fitness scores than all expressed genes, indicating that these genes enhance cell viability (p=0.046; *Figure 3e*). Moreover, *COPB2*, associated with an enhancer mutation in HCT116 cells that is recurrent in three primary CRC tumors, was a dependency gene in these cells (*Figure 3f*). Another recurrent indel gene, *UBE2V2*, scored as a CRC-specific dependency

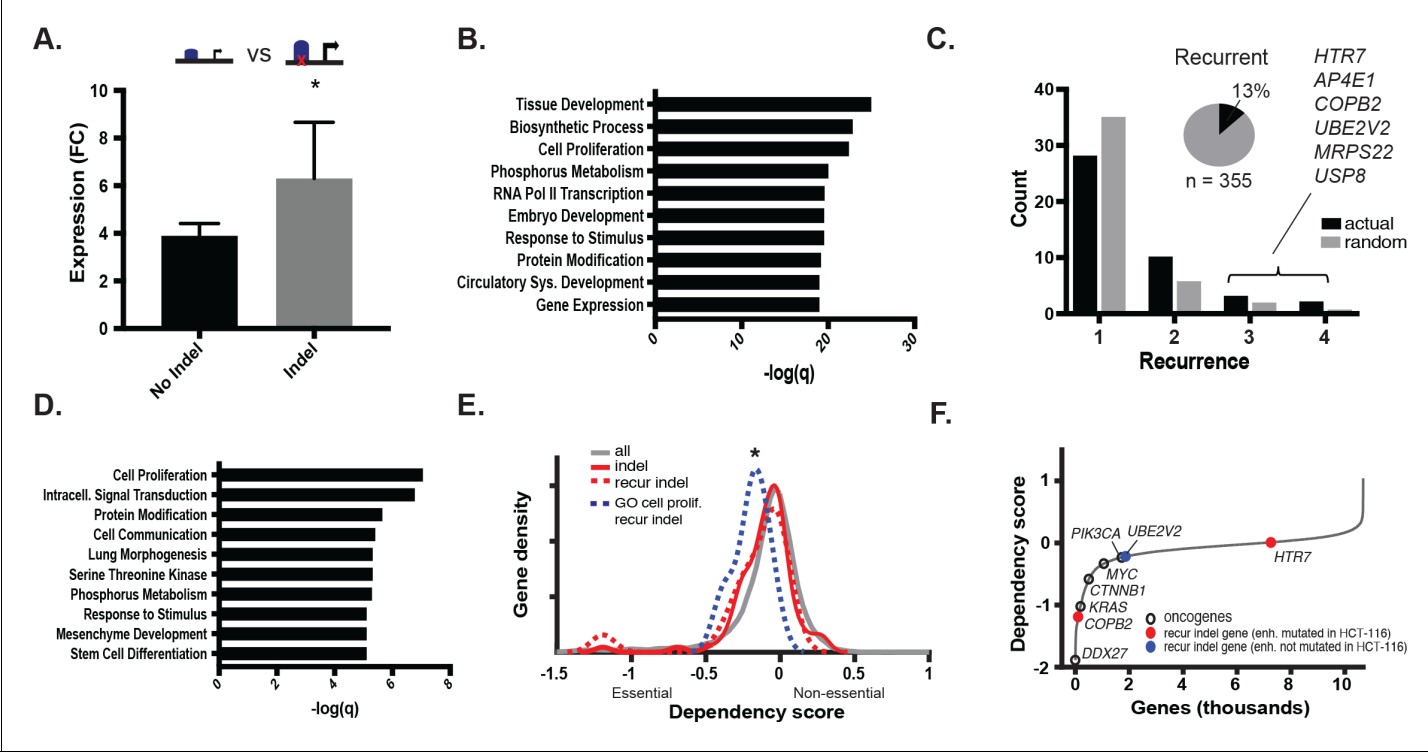

**Figure 3.** MSI enhancer indels impact target gene expression. (**A**) Mean fold-change of expression (CRC/crypts) of genes associated with gained enhancers containing indels (grey; n = 352 sample-gene pairs) versus the same genes in cell lines with gained enhancers but lacking indels (black; n = 2384 sample-gene pairs). *p<0.01, 2-sample t-test (*Figure 3—source data 1*). (**B**) Enriched GO terms associated with enhancer indel genes. (**C**) (black) Distribution of number of TCGA samples with recurrent indel and (grey) average recurrence distribution from random sampling of TCGA indels (p<0.05, χ² test). Listed genes have indels recurrent in three or more TCGA samples. Pie chart shows fraction of enhancer indels that are recurrent in at least one primary CRC tumor (*Figure 3—source data 2*). (**D**) Enriched GO terms associated with recurrent enhancer indel genes. (**E**). Distribution of dependency scores in HCT-116 cells for all expressed genes (grey; n = 13,584), indel genes (red; n = 100), recurrent indel genes (dotted-red; n = 17), and recurrent indel genes with 'Cell Proliferation' annotation (dotted-blue; n = 5). *p<0.05 for 'Cell Proliferation' recurrent indel gene versus all expressed genes, Wilcox ranksum test. (**F**) Genes ranked by dependency scores, showing select oncogenes (black circles), and recurrent indel genes with (red circles) and without (blue circles) enhancer mutation in HCT-116 cells. Data for recurrent indel genes shown in 'C' (*USP8, AP4E1, MRPS22*) were not available.

DOI: https://doi.org/10.7554/eLife.40760.010

The following source data and figure supplement are available for figure 3:

**Source data 1.** Expression of predicted target genes of enhancer indels, recurrence of indels in primary tumors, and enriched GO terms.
DOI: https://doi.org/10.7554/eLife.40760.011

**Source data 2.** Expression of predicted target genes for all gained enhancers with an allele-biased indel, and distribution of recurrent enhancer indels in gained and shared enhancers.
DOI: https://doi.org/10.7554/eLife.40760.012

**Figure supplement 1.** Recurrent MSI enhancer indels are enriched in gained enhancers.
DOI: https://doi.org/10.7554/eLife.40760.013

gene specifically in MSI tumors and not in MSS tumors (*Figure 3—figure supplement 1d*). Together, the data indicate that recurrent enhancer indels likely promote fitness by enhancing cell growth or other processes that confer a selective advantage. We suspect this selective advantage arises over time, as somatic mutations accumulate in the tumor cells due to MMR-deficiency.

## MSI enhancer mutations recruit FOX transcription factors

We set out to identify transcription factors recruited to the indels and potentially responsible for enhancer activation. DNA motif scanning tools revealed forkhead (FOX) sites as most enriched at indels in gained enhancers (*Table 1*). FOX are known pioneer transcription factors in many cell types (*Ang et al., 1993*; *Lupien et al., 2008*; *Iwafuchi-Doi et al., 2016*), and are associated with enhancer

**Table 1.** Top motifs enriched at indels.

The logo displayed is for representative factors from each TF family (FOXC1, IRF1, SOX2, SP1, and EGR1). The total number of indels is 355.

| TF (% of indels) | Motifs |
| --- | --- |
| FOX (20%) | |
| IRF (4%) | |
| SOX (4%) | |
| SP (4%) | |
| EGR (3%) | |

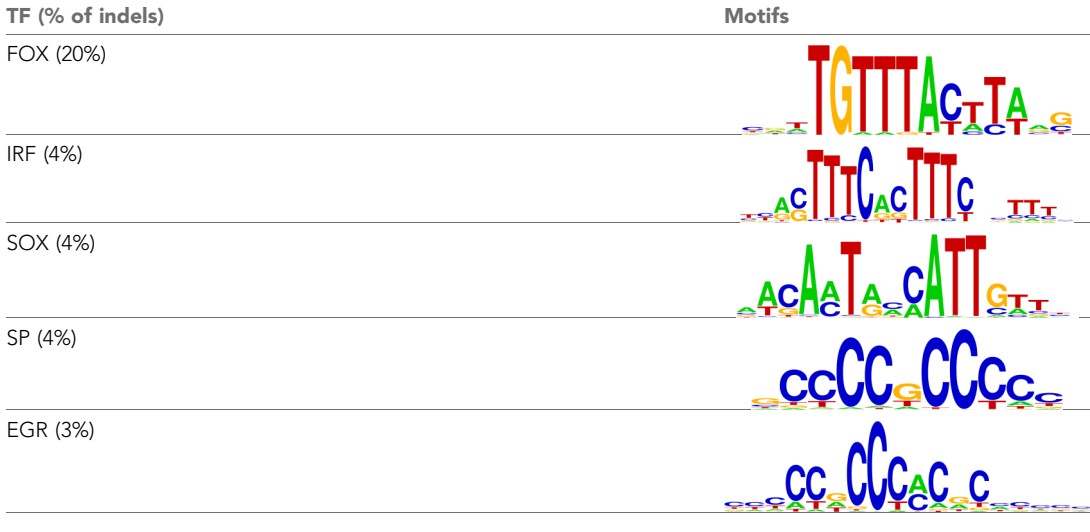

DOI: https://doi.org/10.7554/eLife.40760.016

re-programming (*Pomerantz et al., 2015*; *Roe et al., 2007*). FOX also scored as the top hit among factors predicted to bind with higher affinity to the indel than the wildtype sequence (*Figure 4a–b*, *Figure 4—source data 1*) and was the second most enriched motif (behind CPEB) in the recurrent indels detected in the TCGA CRC cohort. Through integrative analyses of available FOX ChIP-seq datasets (*Yan et al., 2013*) with H3K27ac ChIP-seq data from LoVo cells, we determined that 83% (64/77) of gained enhancer indels show evidence of FOX enrichment by ChIP-seq. An example of an indel locus enriched for both FOXA2 and FOXO3 is shown in *Figure 4c*. Moreover, at the majority of FOX peaks with sufficient coverage at the indel to assess allelic imbalance, FOX ChIP reads support allele-specific FOX binding (*Figure 4d*).

We examined the consensus motifs of FOX TFs expressed in MSI samples. The motifs are all A/T-rich sequences that closely resemble the ones most frequently mutated in MMR-deficient tumors. All share a core 'TGTTT' within the consensus motif (*Figure 4e*, underlined). We hypothesized that if gained enhancers in MSI lines are formed by MMR-signature mutations that instigate FOX recruitment, then MSI samples should contain a higher frequency of gained enhancers at TGTTT(Tn) sequences than MSS samples. We retrieved all TGTTT(Tn) sequences across the genome and queried for gained H3K27ac enrichment. Strikingly, this unbiased analysis revealed that MSI samples had 50% more gained enhancers at TGTTT(Tn) sequences than MSS samples (*Figure 4f*). These observations indicate that gained enhancers arise more frequently at the sequences most prone to mutation as a result of MMR-deficiency. Coupled with the finding that these sites closely resemble the consensus motifs of FOX factors, the computational prediction that the indels increase FOX affinity, and the ChIP-seq results indicating allele-biased FOX binding to the indels, these observations suggest FOX factors mediate enhancer activation at the indels. However, given the degenerate nature of the FOX consensus motifs, additional studies are needed to determine the specific FOX factors responsible for enhancer activation, if they act in cooperation with other factors, and if there is intertumor heterogeneity.

## Induction of MSI phenotype yields enhancer mutations

To functionally test if mutations resulting from MMR deficiency can activate enhancers, we set out to introduce signature MMR mutations in CRC cells in a manner that recapitulates the natural way in which these mutations arise in MSI tumors. If they are indeed functional, introduction of the indels should lead to enhancer activation in combination with a bias in the H3K27ac ChIP-seq signal. We used CRISPR/Cas9 to knockout the *MLH1* gene in Colo-205, a microsatellite stable CRC cell line (workflow summarized in *Figure 5a*). Homozygous *MLH1* knockout was confirmed by sequencing

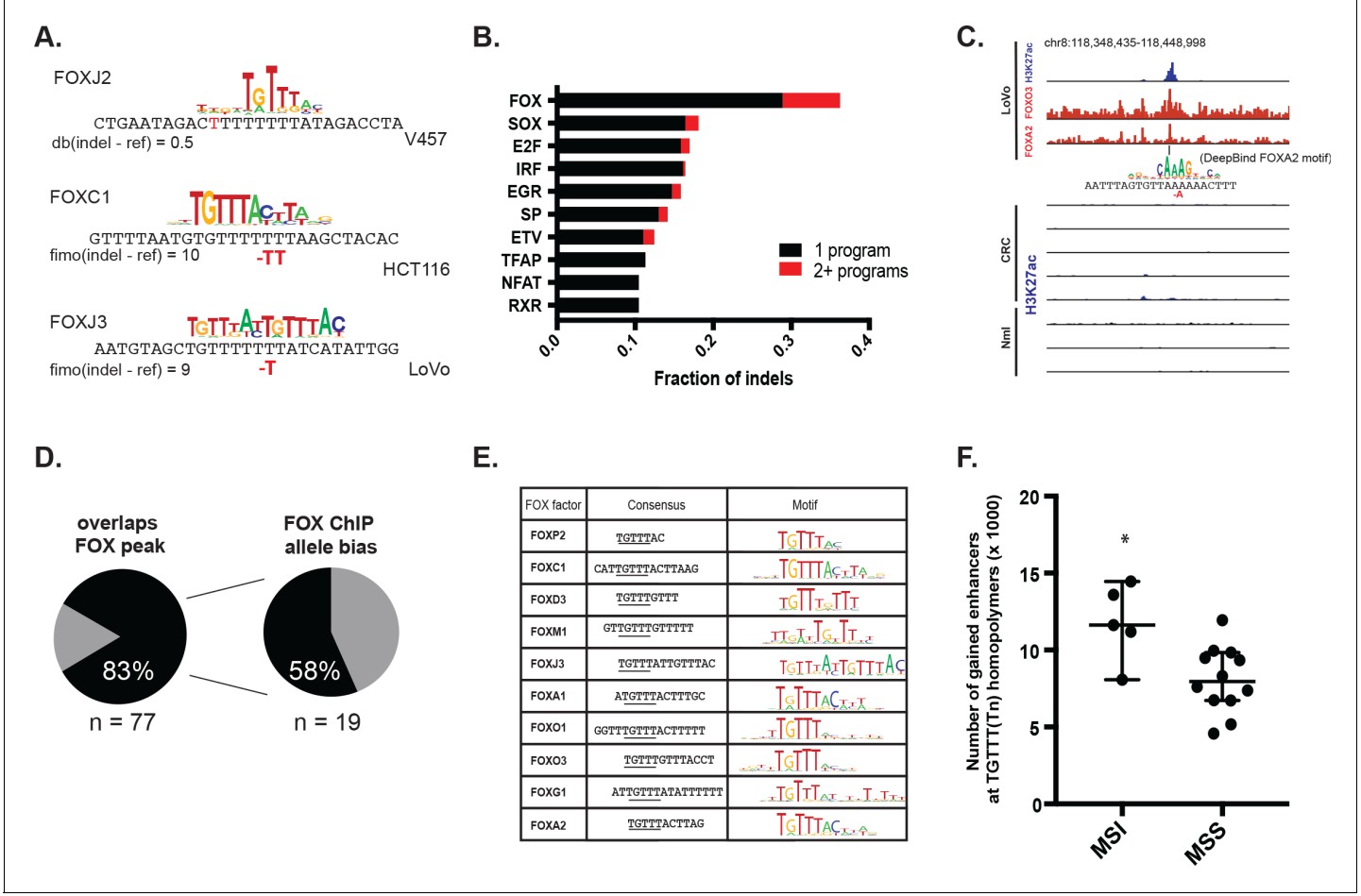

**Figure 4.** MSI enhancer indels recruit FOX factors. (**A**) Examples of FOX motifs at MSI indels, with change in affinity scores. (**B**) Fraction of indels predicted to increase transcription factor affinity by TF family, supported by one (black) or multiple (red) computational programs. (***Figure 4—source data 1***). (**C**) Genome Browser snapshot of MSI enhancer indel overlapping FOXA2 and FOXO3 peaks in LoVo cells. Motif of FOXA2 generated by DeepBind, which predicts increased binding of FOXA2 to the indel alelle, is shown. (**D**) (left) Fraction of LoVo enhancer indels overlapping a FOX factor peak. (right) Fraction of indels in FOX factor peaks and with >3 FOX ChIP read coverage showing allele-bias. (**E**) Motifs of FOX factors expressed in MSI lines and predicted to bind preferentially to indel alleles. (**F**) Number of gained enhancers at TGTTT(Tn) motif in MSI versus MSS lines. *p<0.05, Wilcox ranksum test.

DOI: https://doi.org/10.7554/eLife.40760.014

The following source data is available for figure 4:

**Source data 1.** Transcription factors predicted to bind more favorably to enhancer indels, based on three computational programs.

DOI: https://doi.org/10.7554/eLife.40760.015

and western blot (***Figure 5b***). Following 2.5 months of cell culture, subcloning, and expansion, several $MLH1^{-/-}$ clones tested positive by PCR assay for the MSI phenotype (***Figure 5c***). Two $MLH1^{-/-}$ and two parental wildtype clones were selected for H3K27ac ChIP-seq analysis. Applying our analysis pipeline, we uncovered enhancer indels matching the MMR-associated signature observed in the MSI lines of our panel, namely a high indel rate affecting homopolymers (10x the rate in the WT clones), a bias for short, 1–2 bp deletions, and a bias for poly-(A/T) tracts (***Figure 5d***, ***Figure 5—source data 1***). Most indels in the $MLH1^{-/-}$ cells (1087/1413, or 77%) arose in H3K27ac peaks that were already present at similar levels in parental wildtype clones. This was not unexpected, and likely reflects the switch from low to high mutation rate in open chromatin upon loss of MMR function, as observed previously in ***Figure 2a***. We identified 30 indels in enhancers that showed a 1.5 fold or greater increase in the H3K27ac signal in $MLH1^{-/-}$ cells compared to parental wildtype cells (***Figure 5e*** shows an example). We compared the percentage of mutant-allele reads at the gained enhancer indels to that of the shared enhancer indels. Strikingly, the gained enhancer indels were

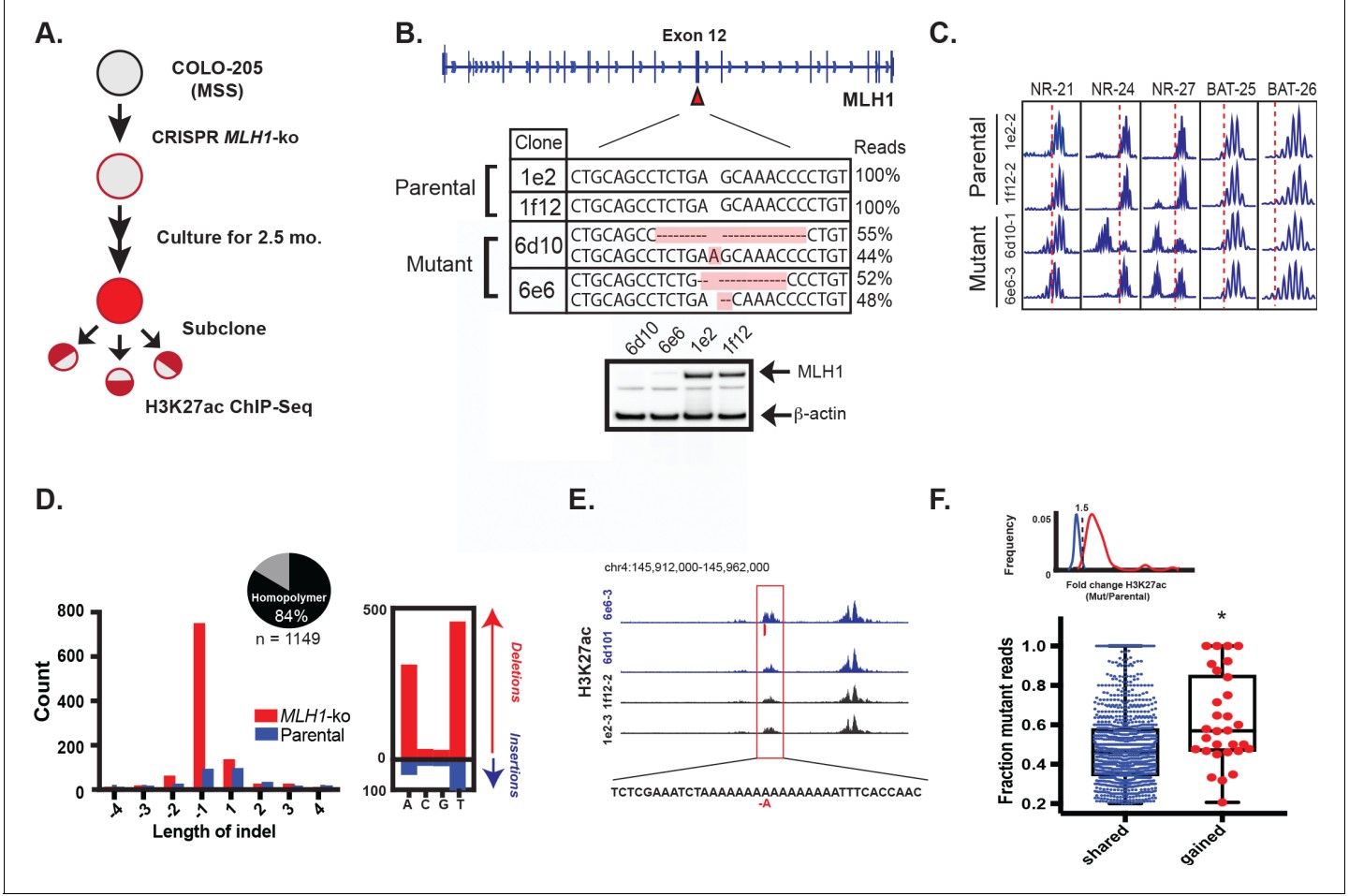

**Figure 5.** MLH1 inactivation induces MSI and yields enhancer mutations. (**A**) Overview of MSI induction experiment. (**B**) CRISPR-mediated knockout of *MLH1*. Mutations highlighted in orange. (bottom) Western blot analysis of MLH1 in *MLH1* wildtype and mutant clones. Beta-actin is shown as a loading control. (**C**) PCR assay of five MSI markers in *MLH1*$^{+/+}$ and *MLH1*$^{-/-}$ clones. (**D**) Distribution of enhancer indel lengths. Pie chart represents fraction of indels affecting homopolymers. (right) Count of homopolymer insertion (blue) and deletion (red) mutations by mononucleotide repeat. (**E**) Genome browser snapshot of indel (red bar) associated with increase in H3K27ac signal. Y-axis scales are all 0 to 90. (**F**) Mutant allele fraction at indels in shared and gained H3K27ac peaks. (top) Density of H3K27ac signal fold change (*MLH1*$^{-/-}$/*MLH1*$^{+/+}$) for shared (blue) and gained (red) peaks in *MLH1*$^{-/-}$ cells, relative to *MLH1*$^{+/+}$ cells. (bottom) Dot plot of the mutant allele fraction for indels in shared (n = 1087) and gained (n = 30) peaks. *p<0.001, Wilcox ranksum test (*Figure 5—source data 1*).

DOI: https://doi.org/10.7554/eLife.40760.017

The following source data is available for figure 5:

**Source data 1.** Enhancer indels detected from H3K27ac ChIP-seq of *MLH1*$^{-/-}$ and *MLH1*$^{+/+}$ cells, cultured for 2.5 months.

DOI: https://doi.org/10.7554/eLife.40760.018

more often allele-biased (*Figure 5f*) suggesting the basis of enhancer activation was acquisition of the indel. We further note that the frequency of indel-gained enhancer events detected in this experiment is likely to be lower than in naturally-derived MSI tumors, since the CRISPR-engineered cells spent a limited time in culture and were grown under conditions that do not recapitulate selective pressures of the tumor microenvironment.

## MSI enhancer indels are propagated in tumors

To test if loss of *MLH1* induces indels that activate enhancers in vivo, we introduced *MLH1* knockout (clone 6e6-3) and wildtype (clone 1f12-2) cells into mice via intrasplenic injection (workflow is summarized in *Figure 6a*). In this assay, tumor cells typically form clonal liver metastases, but are also known to form peritoneal tumors (*Lee et al., 2014*). To simulate different micro-environmental

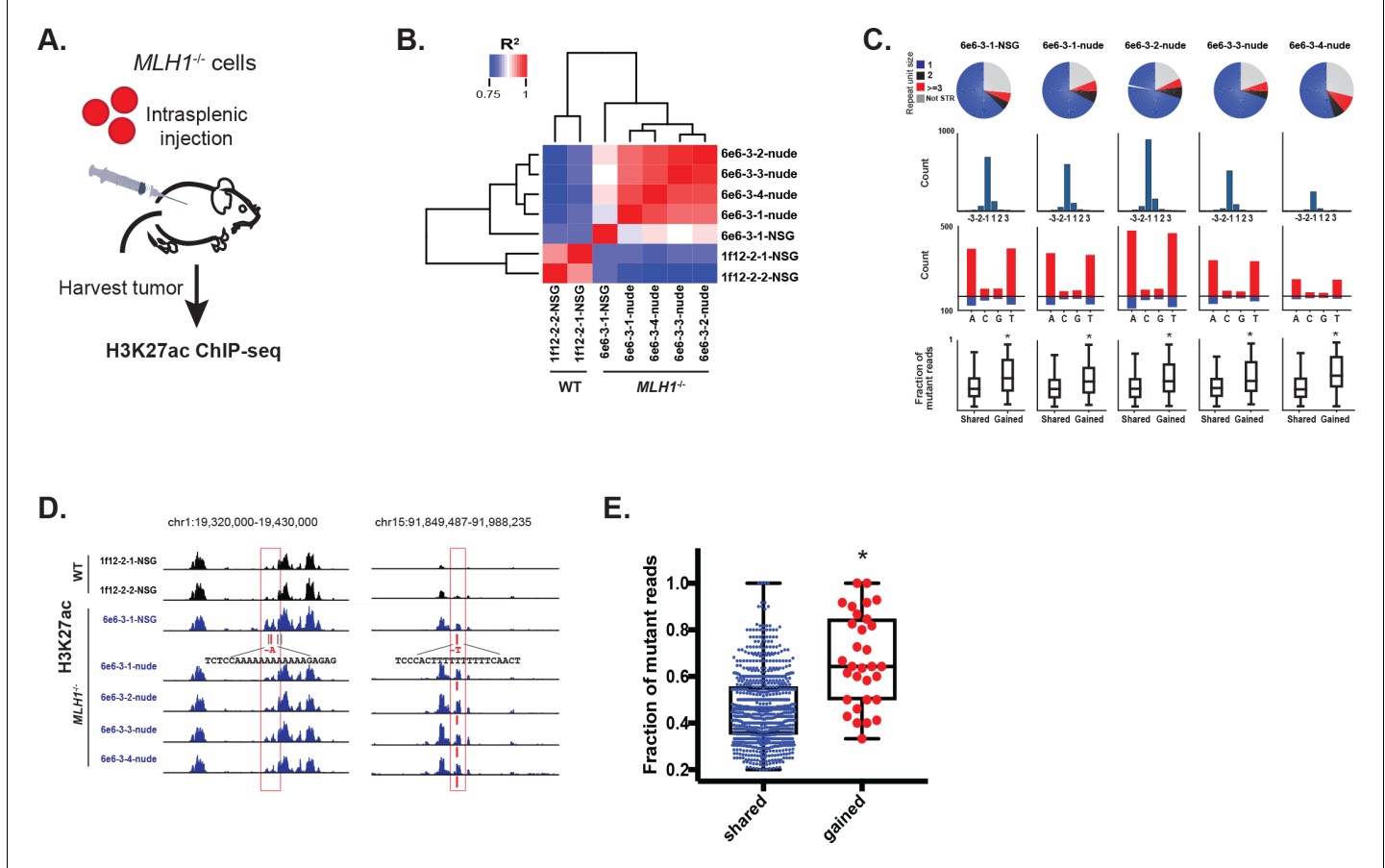

**Figure 6.** MSI enhancer indels are propagated in tumors. (A) Overview of mouse tumor formation assay. (B) Heatmap of $R^2$ values from Pearson correlation of H3K27ac signal at enhancers. (C) (top) Pie charts show distribution of types of mutations (mono-, di-, tri-nucleotide or higher, and non-short tandem repeat). (2nd row) Histogram of indel sizes. (3rd row) Bar plots of homopolymer indel frequency by mononucleotide repeat. (bottom) Boxplots of the mutant allele fraction distribution for indels in gained peaks and peaks shared with wildtype tumors. *p<0.001, Wilcox ranksum test. (D) Genome Browser snapshots of indels (red bars) detected in MLH1-/- tumors and associated with increased H3K27ac signal (left, Y-axis scales are all 0–156; right, Y-axis scales are all 0–95). (E) Dot plot of the mutant allele fraction for homopolymer indels in shared (n = 894) peaks and those correlating with gained H3K27ac enrichment (n = 31). *p<0.001, Wilcox ranksum test. See *Figure 6—source data 1*.

DOI: https://doi.org/10.7554/eLife.40760.019

The following source data is available for figure 6:

**Source data 1.** Enhancer indels detected from H3K27ac ChIP-seq of tumors derived from *MLH1-/-* and *MLH1+/+* cells introduced through intrasplenic injection into mice.
DOI: https://doi.org/10.7554/eLife.40760.020

pressures, we used two mouse strains (nude and NSG) with different levels of immune competence. Three months post-injection, five liver tumors were harvested from mice seeded with *MLH1-/-* cells (four tumors from one nude mouse and one tumor from one NSG mouse). Two peritoneal tumors were harvested from one NSG mouse injected with *MLH1+/+* cells. We then performed H3K27ac ChIP-seq profiling of all seven tumors. Unsupervised cluster analysis of the enhancer landscapes separated *MLH1+/+* from *MLH1-/-* tumors, and there was remarkable consistency among tumors with matched genotypes (*Figure 6b*). Analysis of the mutations detected from H3K27ac ChIP-seq of the *MLH1-/-* tumors again revealed predominantly small 1–2 bp deletions in mononucleotide tracts of A/ T repeats (*Figure 6c*, *Figure 6—source data 1*). We also noted a shift in the proportion of non-homopolymer mutations, including indels in larger tandem repeats and in non-tandem repeat regions. In each *MLH1-/-* tumor, 5–10% of the indels were located in enhancers that showed 1.5 fold or greater increase in the H3K27ac signal relative to *MLH1+/+* tumors. Consistent with an enhancer-activating role, indels in gained peaks again showed higher mutant allele fractions compared to

indels in H3K27ac peaks already present at similar levels in the parental cells (*Figure 6c*, bottom). To account for potential site-specific differences, we looked not only for enhancer activating mutations in liver tumors versus the peritoneal tumors, but also between individual liver tumors. We identified 31 indels in *MLH1*$^{-/-}$ tumors that showed a perfect correlation with an increase in H3K27ac (see *Figure 6d* for examples). Strikingly, these indels also showed high mutant allele fractions (*Figure 6e*). Interestingly, amongst the predicted target genes of the gained enhancer indels, *OGDH* was found as an MSI CRC-specific dependency gene. Together, these data provide further functional support that MSI signature mutations activate enhancers in vivo.

## Discussion

The search for functional mutations in non-coding regions of tumor genomes is an area of intense investigation. Beyond those affecting *TERT*, *ER*, *FOXA1* and others, regulatory mutations are presumed rare and many are considered passengers (*Fredriksson et al., 2014*; *Cuykendall et al., 2017*). Consistent with this supposition, the majority of samples in our CRC cohort showed a low enhancer mutation rate. However, a clear exception are CRC tumors whose mutation rates across the genome are extraordinarily high, like the MSI subtype. In MSI CRC samples, we detected a large number of indel mutations that correlate with higher levels of H3K27ac in tumor cells relative to normal colon cells. In addition to showing allele-bias, several lines of evidence indicate these indels functionally activate enhancers and are not merely random passengers. First, they show signatures of positive selection. Second, their target genes, in addition to being enriched for cancer-related functions, are more highly expressed when both the gained enhancer and indel are present compared to when only the gained enhancer is present. Third, a higher-than-expected number of the enhancer indels are recurrent across primary CRC tumor samples. Fourth, the indels are predicted to enhance FOX binding and an unbiased genome-wide scan of the core FOX motif sequence TGTTT (Tn) showed 50% more gained H3K27ac signals at this motif in MSI samples than MSS samples. Lastly, introduction of the indels via CRISPR-Cas9 inactivation of *MLH1* - thereby mimicking the natural way in which these mutations occur - led to allele-biased enhancer activation and selection of enhancers both in cell culture and in full blown tumors, suggesting that the basis of enhancer activation in these cells was acquisition of the mutation. Based on these findings, we conclude that in CRC, deficiencies in mismatch repair lead to the appearance of indels that are recognized by FOX factors, which turn these sites into functional cis-regulatory elements. It has long been established that signature MMR-mutations, when they arise in coding regions, can impact gene splicing or shift the reading frame. Indel mutations that induce a frameshift in *TGFBR2* are a notable example (*Markowitz et al., 1995*). Our studies here expand the repertoire of functional MSI-type mutations to non-coding regions, and offer a new strategy for identifying functional mutations in regulatory elements.

There are notable differences between the functional regulatory mutations described here and those previously linked to *TERT*, *ER*, *TAL1*, *MYC*, and *FOXA1*. The latter are considered tumor-initiating events and activate *bona fide* drivers of oncogenesis. In contrast, most of the enhancer mutations reported here are a consequence of MMR-inactivation and confer a fitness advantage beyond that conferred by mutations in the canonical CRC driver genes. In support of this model, several of the genes with mutated enhancers are modest dependency genes linked to proliferative functions. The key finding is that the prevalence of these mutations in MSI CRC is especially high, suggesting that their cumulative effect on growth is substantial. Given their prevalence and widespread effects on gene expression, MSI enhancer mutations could be considered 'reprogrammers' of cell identity. We further note, given that MSI is a continuously evolving phenotype, that enhancer mutations due to MMR-deficiencies could play an important role in tumor evolution and the emergence of drug resistance. Furthermore, as the MSI subset of CRC is a particularly good candidate for immunotherapy, it would be worth investigating if enhancer mutations instigate expression of PD-1 pathway genes or neoantigen-producing genes that contribute to tumor immunogenicity and may therefore be exploited for refining response prediction.

# Materials and methods

**Key resources table**

| Reagent type or resource | Designation | Source or reference | Identifiers | Additional information |
|---|---|---|---|---|
| Cell line (*Homo-sapiens*) | LoVo | ATCC | RRID:CVCL_0399 | |
| Cell line (*Homo-sapiens*) | Colo-205 | ATCC | RRID:CVCL_0218 | |
| Cell line (*Homo-sapiens*) | 6d10-1 | This paper | | Derived from Colo-205. These cells were CRISPR-edited to knock out *MLH1* (to get $MLH1^{-/-}$ cells), cultured for 2.5 months, then subcloned. |
| Cell line (*Homo-sapiens*) | 6e6-3 | This paper | | Derived from Colo-205. These cells were CRISPR-edited to knock out *MLH1* (to get $MLH1^{-/-}$ cells), cultured for 2.5 months, then subcloned. |
| Cell line (*Homo-sapiens*) | 1e2-3 | This paper | | Derived from Colo-205. These cells were from the same nucleofected pool as *MLH1*-ko clones but were not edited (are $MLH1^{+/+}$ cells), cultured for 2.5 months, then subcloned. |
| Cell line (*Homo-sapiens*) | 1f12-2 | This paper | | Derived from Colo-205. These cells were from the same nucleofected pool as *MLH1*-ko clones but were not edited (are $MLH1^{+/+}$ cells), cultured for 2.5 months, then subcloned. |
| Antibody | anti-MLH1 (mouse monoclonal) | BD Pharmingen | RRID:AB_394040 | (1:250) |
| Antibody | anti-beta actin (rabbit polyclonal) | Abcam | RRID:AB_2305186 | (1:1000) |
| Antibody | anti-rabbit IgG (H + L) (goat polyclonal) | ThermoFisher | RRID:AB_228341 | (1:20000) |
| Antibody | anti-H3K27ac (rabbit polyclonal) | Abcam | RRID:AB_2118291 | (1:50) |
| Software, algorithm | indel calling pipeline | This paper | | Stores scripts used to call indel mutations from H3K27ac ChIP-seq data. Available at https://github.com/scacherilab/indel_calling_pipeline (*Saiakhova, 2019*; copy archived at https://github.com/elifesciences-publications/indel_calling_pipeline). |

## ChIP-seq

H3K27ac and H3K4me1 ChIP-seq datasets from CRC cell lines and primary tumors are previously described in our previous publications (*Cohen et al., 2017*; *Akhtar-Zaidi et al., 2012*) and available in GEO (Accession numbers GSE36401 and GSE77737). H3K27ac ChIP-seq was performed on LoVo, Colo-205, and $MLH1^{-/-}$ Colo-205 cells as previously described. ChIP data processing, alignment, peak-calling, and identification of differentially enriched-peaks relative to normal colonic crypts were done as previously described.

## Detection of enhancer mutations

H3K27ac ChIP seq reads were aligned to the human genome (hg19) with Bowtie2. Reads were realigned around regions with evidence of indels (>1 indel supporting read) using GATK v.2–2-

gec30cee. A custom FASTA file was created incorporating the candidate indel sequences and their flanking regions such that the length of the flanking region equals the length of the longest aligned read. Any read that aligned perfectly to both reference and custom indel genomes was discarded. Samtools 1.2 was used to generate a multi-way pileup output file for each filtered BAM. Indels were called using VarScan.v2.4.0 from the pileup output, requiring at least 10X coverage and 20% of reads supporting the indel. To exclude possible germline variants, indels matching dbSNPs from the 1000 Genomes Project and/or SNP142 indels were filtered out.

Candidate enhancer-activating mutations were prioritized based on whether they correlate with gained H3K27ac enrichment. Binary matrices of indels and RPKM matrices were constructed for all H3K27ac peaks. A peak was reported as correlating with an indel if the following conditions were met:

1. Peak RPKM for the sample the peak was called in is at least 2.
2. Minimum peak RPKM for samples with indels in the peak is greater than the maximum RPKM for the samples with no reported indels in the same peak.
3. No reads support the indel in samples not called to have the indel.

Indels were filtered if they overlapped artifact regions in the consensus Blacklist (https://sites.google.com/site/anshulkundaje/projects/blacklists). An empirical approach was used to identify indel calls that are likely alignment errors. For each indel, the sequence 50 bp upstream and downstream was aligned to the human genome (Blat, from UCSC Genome Browser). The reference allele was replaced with the indel allele, to simulate the alignment of indel-supporting ChIP reads. Indels whose second-highest alignment score was >50 indicated potential alignment error and were discarded.

Indels with imbalanced read distributions favoring the indel allele were prioritized, as this suggests a scenario whereby the enhancer signal and indel co-occur on the same allele. Bias in the number of reads supporting the mutation was quantified by the complement of the cumulative binomial distribution (upper-tail), with a probability of success of 0.5. Multiple test correction was performed using the Benjamini-Hochberg method controlling the FDR at 0.2.

## Correlation of H3K27ac enrichment and indel rate

Chromosome 14 was split into 537 bins of 200 kb. For each bin, the median H3K27ac signal from the V410 or HCT-116 bigWig was retrieved. Indel calls from either MSS samples or MSI samples in the TCGA cohort were pooled and indels were counted in each bin. The H3K27ac signal and indel rate were quantile-normalized and smoothed. This analysis was repeated for chromosomes 1 through 22 and chromosome X using 0.5 Mb genomic bins, to obtain genome-wide correlations of H3K27ac and mutation density. For comparisons of the indel rate in gained and shared enhancers, the indel rate was defined as the number of indels divided by the length of the enhancer peak, and plotted as a function of the peak's median H3K27ac bigWig signal. Regression analysis using a generalized linear model (binomial distribution) was performed to get the significance of gained enhancer status as a predictor for indel rate.

## Prediction of indel gained enhancer gene targets

GREAT (version 3.3.0) was used to pair gene targets with enhancers at each locus ('basal plus extension' setting), and to get the distance to the nearest transcription start site. For each predicted target gene, the (quantile-normalized) expression from microarray data was retrieved for the line harboring the indel gained enhancer, and lines harboring gained enhancers without indels at the locus. Fold change relative to the median expression of that gene in five normal crypt lines was calculated. Enriched gene ontology terms were obtained by inputting (expressed) GREAT genes to the GSEA web tool to compute overlap with the 'C5', or GO gene sets in the MSigDB database. Data from gene knock-down studies in HCT-116 cells were downloaded from the DepMap project data portal (*McFarland et al., 2018*).

## Mutation calling from TCGA COAD and in-house WGS data

Whole genome sequence reads aligned to GRCh37 (.bam files) of tumor and normal matched pairs were downloaded from the Cancer Genomics Hub (UCSC) using the GeneTorrent tool, and re-processed according to GATK best practices. First, base quality scores were reverted (Picard v.1.104

RevertSam), but duplicate and alignment information was retained. Next, the alignment of reads around indels was redone and then base quality scores were re-calibrated using GATK IndelRealinger and BaseRecalibrator (version 3.2–2-gec30cee), respectively. Somatic substitutions and indels were called using two mutation callers, MuTect2 (v3.5–0-g3628e4) and Varscan2 (v2.3.9) using default settings, and the intersection of the calls was used for further analysis. Variants in common dbSNP and in 1000 Genomes were filtered out as likely germline events. Significance of the recurrence distribution was determined by randomly picking indels (same number as actual recurrent indel set) from the binary matrix and finding the average recurrence distribution of those random indels.

## Transcription factor motif analysis

Ten (non-homopolymer) base pairs of the reference genome (hg19) upstream and downstream of the indel were retrieved and used to flank either the reference allele or the indel allele. Both indel and reference sequences were input to three computational programs which score the affinity of their interaction with transcription factors: Deepbind (*Alipanahi et al., 2015*), FIMO (*Grant et al., 2011*) with the HOCOMOCO v.10 motif database, and Footprint (*Sebastian and Contreras-Moreira, 2014*), with the Human-TF v2.0 (*Jolma et al., 2015*) motif database. For each program, a 'binding score' was retrieved for each TF-sequence pair, and compared between reference and indel alleles. A prediction of increased binding was counted if binding score$_{indel}$ >binding score$_{reference}$ and the predicted TF is expressed in the cell line harboring the indel. TF predictions were collapsed by TF family. Unless otherwise noted, logos were created from HOCOMOCO v.10 PWM's. FOX ChIP-seq data from the LoVo cell line (*Yan et al., 2013*) was downloaded (GEO GSE51142). Reads were re-aligned to the human reference genome (hg19) using Bowtie2 to generate BAM files. Peaks were called using macs1.4. Reads at indel loci were retrieved using samtools view command. To calculate allele bias, indels with coverage of 4 or more FOX ChIP-seq reads were selected, and the complement of the cumulative binomial distribution, with a probability of success of 0.5, was used. A false discovery rate of 0.2 was used, such that tests with p-value<0.11 were deemed significant.

## CRISPR-mediated *MLH1*-knockout in Colo-205 cells

*MLH1* knockout was performed in collaboration with the Genome Engineering and iPSC Center (GEiC) at Washington University. Guide RNAs were designed to target a conserved exon (exon 12) of the *MLH1* gene. The gRNAs and Cas9 were nucleofected into Colo-205 (MSS) cells, and then individual lines were subcloned. CRISPR-induced indels were verified by targeted NGS. WT controls were unmodified clones from the same nucleofected pool of cells.

## Cell culture

*MLH1* knockout and parental WT lines were cultured for 2.5 months in RPMI-1640 media supplemented with 10% fetal bovine serum (ThermoFisher 11875093) to accumulate mutations. Monoclonal cells were then derived by limiting dilution, assayed for instability at MSI markers, and then expanded for 3–4 weeks before using in H3K27ac ChIP-seq experiment. Cell lines were checked for mycoplasma contamination and found to be negative.

## PCR assay for microsatellite instability

Genomic DNA was extracted and PCR-amplified at five MSI markers (*Buhard et al., 2004*; *Buhard et al., 2006*). PCR primers are from *Buhard et al. (2006)*, with one primer in each pair labeled with a fluorescent dye for combined analysis. PCR products were run on a Genetic Analyzer ABI 3730 to produce fragment profiles, which were visualized using Peak Scanner version 1.0 (ThermoFisher).

## Mice

Two 6 week old female Nod-Scid IL-2Rg$^{-/-}$ (NSG) mice and two nude mice were each injected intra-splenically with $10^6$ (6e6-3 cells or 1f12-2) cells. After 3 months, mice were sacrificed. Tumors were harvested and homogenized for H3K27ac ChIP-seq experiment. All mouse experiments were done according to Institutional Animal Care and Use Committees (IACUC) guidelines.

## Acknowledgements

We thank Grace Lee from Dr. David Wald's group for experimental assistance preparing the mice for surgery. We thank the Case Western Reserve University Genomics Core for their support with ChIP-sequencing and genotyping, and the High Performance Computing Cluster for their computational infrastructure. We thank Samuel Li and Sneha Grandhi from Dr. Thomas LaFramboise's lab for help downloading WGS data. We thank Shiyi Yin from Dr. Ann Harris' lab and Chen Weng from Dr. Fulai Jin's lab for helpful discussions. We thank Dr. Thomas LaFramboise, Dr. Paul Tesar and Dr. Goutham Narla for valuable advice. We thank Monica Sentmanat and the Genome Engineering and iPSC Center at Washington University at St. Louis for generating the $MLH1^{-/-}$ clones. Finally, we are grateful to Dan Weidenthal for his generous contribution.

## Additional information

### Funding

| Funder | Grant reference number | Author |
|---|---|---|
| National Institutes of Health | TL1 TR000441 | Stevephen Hung |
| National Institutes of Health | T32 GM007250 | Stevephen Hung |
| Case Western Reserve University | P50 CA150964 | Sanford Markowitz |
| National Institutes of Health | R01CA160356 | Peter C Scacheri |
| National Institutes of Health | R01CA193677 | Peter C Scacheri |
| National Institutes of Health | R01CA204279 | Peter C Scacheri |
| National Institutes of Health | R01CA143237 | Peter C Scacheri |

The funders had no role in study design, data collection and interpretation, or the decision to submit the work for publication.

### Author contributions

Stevephen Hung, Conceptualization, Software, Formal analysis, Validation, Investigation, Visualization, Methodology, Writing—original draft, Project administration, Writing—review and editing; Alina Saiakhova, Conceptualization, Resources, Data curation, Software, Formal analysis, Visualization, Methodology; Zachary J Faber, Validation, Investigation, Visualization, Writing—original draft; Cynthia F Bartels, Ian Bayles, Evelyn Ojo, Ruifu Liu, Validation, Investigation; Devin Neu, Formal analysis, Validation, Investigation; Ellen S Hong, Validation and Investigation; W Dean Pontius, Formal analysis, Visualization; Andrew R Morton, Visualization, Methodology; Matthew F Kalady, Resources and Methodology; David N Wald, Resources, Methodology; Sanford Markowitz, Resources, Supervision, Funding acquisition; Peter C Scacheri, Conceptualization, Resources, Data curation, Formal analysis, Supervision, Funding acquisition, Visualization, Methodology, Writing—original draft, Project administration, Writing—review and editing

### Author ORCIDs

Stevephen Hung http://orcid.org/0000-0001-7067-9125
Cynthia F Bartels https://orcid.org/0000-0001-8790-4288
W Dean Pontius https://orcid.org/0000-0002-3883-0048
Matthew F Kalady https://orcid.org/0000-0002-2114-114X
Peter C Scacheri http://orcid.org/0000-0002-7629-918X

### Ethics

Animal experimentation: Mouse studies were performed in accordance with protocols approved by the Case Western Reserve University Institutional Animal Care and Use Committee (Protocol Number: 2014-0088), and in collaboration with the Athymic Animal and Xenograft Core of the Case Comprehensive Cancer Center.

Decision letter and Author response
Decision letter https://doi.org/10.7554/eLife.40760.031
Author response https://doi.org/10.7554/eLife.40760.032

## Additional files

### Supplementary files

• Transparent reporting form
DOI: https://doi.org/10.7554/eLife.40760.023

### Data availability

Data are available via the NCBI GEO repository (accession number GSE126188).

The following dataset was generated:

| Author(s) | Year | Dataset title | Dataset URL | Database and Identifier |
|---|---|---|---|---|
| Stevephen Hung, Alina Saiakhova, Zachary J Faber, Cynthia F Bartels | 2019 | Mismatch repair-signature mutations activate gene enhancers across human colorectal cancer epigenomes | https://www.ncbi.nlm.nih.gov/geo/query/acc.cgi?acc=GSE126188 | NCBI Gene Expression Omnibus, GSE126188 |

The following previously published datasets were used:

| Author(s) | Year | Dataset title | Dataset URL | Database and Identifier |
|---|---|---|---|---|
| Cohen AJ, Saiakhova A, Corradin O, Luppino JM, Lovrenert K, Bartels CF, Morrow JJ, Mack SC, Dhillon G, Beard L, Myeroff L, Kalady MF, Willis J, Bradner JE, Keri RA, Berger NA, Pruett-Miller SM, Markowitz SD | 2017 | Hotspots of aberrant enhancer activity punctuate the colorectal cancer epigenome | https://www.ncbi.nlm.nih.gov/geo/query/acc.cgi?acc=GSE77737 | NCBI Gene Expression Omnibus, GSE77737 |
| Akhtar-Zaidi B, Cowper-Sal-lari R, Corradin O, Saiakhova A, Bartels CF, Balasubramanian D, Myeroff L, Lutterbaugh J, Jarrar A, Kalady MF, Willis J, Moore JH, Tesar PJ, LaFramboise T, Markowitz S, Lupien M, Scacheri PC | 2012 | Epigenomic enhancer profiling defines a signature of colon cancer | https://www.ncbi.nlm.nih.gov/geo/query/acc.cgi?acc=GSE36401 | NCBI Gene Expression Omnibus, GSE36401 |
| Yan J, Enge M, Whitington T, Dave K, Liu J, Sur I | 2013 | Transcription Factor Binding in Human Cells Occurs in Dense Clusters Formed around Cohesin Anchor Sites | https://www.ncbi.nlm.nih.gov/geo/query/acc.cgi?acc=GSE49402 | NCBI Gene Expression Omnibus, GSE49402 |

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
