## [Decision Letter]

Thank you for submitting your article "Mismatch-repair signature mutations activate gene enhancers across colorectal cancer epigenomes" for consideration by *eLife*. Your article has been reviewed by two peer reviewers, and the evaluation has been overseen by a Reviewing Editor and Aviv Regev as the Senior Editor. The following individual involved in review of your submission has agreed to reveal his identity: Charles Y Lin (Reviewer #3).

The reviewers have discussed the reviews with one another and the Reviewing Editor has drafted this decision to help you prepare a revised submission.

Summary:

Studies of mutations in regulatory regions in cancer have lagged substantially behind those of coding mutations, due to both limited ability to interpret them, and lack of consensus in the field regarding their importance ("driver" vs. "passenger"). The manuscript describes new computational and experimental methods for the analysis of mutation at cis-regulatory regions, including analytical tools and the generation of engineered cell lines for the investigation of MMR on establishing enhancer indels. Combined, these resources are valuable tools to understand the role of non-coding region mutations on oncogenesis.

In application to colorectal cancer, the authors show that microsatellite instable (MSI) tumors have a higher rate of enhancer indels leading to gain of enhancer function and that gained enhancers undergo positive selection. The authors’ schema to identify gained enhancers using allelic imbalance and presence only in CRC specific gained enhancers validates well by sanger sequencing. They present an overall model where defects in mismatch repair (MMR) lead to widespread indels at both open and closed regions of chromatin. Gained enhancers show evidence of positive selection as indicated by maintained allelic imbalance, increased expression of putative target genes. The authors conclude with a thought provoking finding that "in CRC, deficiencies in mismatch repair lead to the appearance of indels that are recognized by FOX factors, which turn these sites into functional-regulatory elements".

Essential revisions:

The study claims are provocative and interesting, and the methods are important resources for studying regulatory mutations. However, as explained in these major comments, the authors did not provide definitive evidence for these claims, and several analyses and explanations are required in the revision to assess enhancer activity, gene regulation and TF binding prior to publication at *eLife*.

1) Strengthening of the oncogenic role of the phenomenon.

It is logical to assume that any enhancer activating the expression of an oncogene will undergo positive selection and also be considered an oncogenic event. Although the authors clearly demonstrate the positive selection aspect, we would like them to consider ways to bolster the oncogenic part of the argument.

• Do recurrent indels actually confer an oncogenic advantage? Is there any evidence of recurrent indels at the pathway level (indels at different loci, but converging on the same pathways)?

• Are genes associated with recurrent indels more likely to be tumor dependencies than non-recurrent genes? For example, in the Dependency Map data, USP8 seems to be a good dependency, but MSX2 is not.

• If data is available, is there any evidence of recurrent indels undergoing clonal selection?

• If most indels are potentially regulated by FOX TFs, would it be reasonable to hypothesize that MSI CRCs are more susceptible to KD of FOX TFs than MSS CRCs? Such a result would go a long way to supporting the claim that "While some of the enhancer indel targets could be oncogenes, most are genes with cancer-related functions that likely provide a selective growth advantage."

We appreciate that some of these queries would be more straightforward to address than others, but expect the authors to address at least some, which can be answered with existing data.

2) More comprehensive demonstration of the statistical associations.

In two key places the authors make statements about association (or lack thereof) to chromatin state, but either do not show the data or choose a partial view:

• The authors mentioned that enhancer regions overlapped with a large percentage of H3K4me1 ChIP-seq, but the data has not been shown. The authors should add the data supporting for this claim.

• On Figure 2A, the authors claimed that there were no associations between mutation rate and H3K27ac regions in MSI samples. Given that the image on Figure 2A shows the distribution of H3K27ac peaks and mutations specific to chromosome 14, the authors' conclusion would be better supported with a genome-wide association analysis of H3K27ac levels and indels. This should be readily rectified.

3) Explanation of the analysis of tumors from mice from different locations.

In Figure 6, the authors injected CRISPR-Cas9 induced MMR CRC cells, and control parental lines, into the portal vein of mice to allow for tumor development and downstream epigenomic/genomic analysis. The authors described that CRISPR-Cas9 induced MMR CRC tumors were harvested from liver, while control parental tumors were harvested from peritoneum. However the authors did not justify why tumors at different locations were chosen for the analysis. In addition, the authors did not investigate possible changes on liver tumor development or tumor histology that could indicate changes to oncogenesis induced by increased enhance activity or mutations. A more in depth analysis of these points could provide further support to the authors final hypothesis.

---

## [Author Response]

Essential revisions:1) Strengthening of the oncogenic role of the phenomenon.It is logical to assume that any enhancer activating the expression of an oncogene will undergo positive selection and also be considered an oncogenic event. Although the authors clearly demonstrate the positive selection aspect, we would like them to consider ways to bolster the oncogenic part of the argument.• Do recurrent indels actually confer an oncogenic advantage? Is there any evidence of recurrent indels at the pathway level (indels at different loci, but converging on the same pathways)?• Are genes associated with recurrent indels more likely to be tumor dependencies than non-recurrent genes? For example, in the Dependency Map data, USP8 seems to be a good dependency, but MSX2 is not.• If data is available, is there any evidence of recurrent indels undergoing clonal selection?• If most indels are potentially regulated by FOX TFs, would it be reasonable to hypothesize that MSI CRCs are more susceptible to KD of FOX TFs than MSS CRCs? Such a result would go a long way to supporting the claim that "While some of the enhancer indel targets could be oncogenes, most are genes with cancer-related functions that likely provide a selective growth advantage."We appreciate that some of these queries would be more straightforward to address than others, but expect the authors to address at least some, which can be answered with existing data.

These are excellent suggestions. In response, we mined for Gene Ontology terms enriched among the recurrent indel-associated genes. The top-enriched GO terms reflect known cancer hallmarks, including cell proliferation, signal transduction, and regulation of cell death (Figure 3D). Per the reviewers’ suggestion, we also cross referenced the recurrent indel genes with dependency data from Dep Map. As a set, the recurrent indel-genes are not enriched for dependency genes (P < 0.15), likely owing to power. However, there are several dependency genes within the set, including many genes involved in “cell proliferation”. Of particular interest, *COPB2*, associated with one of the most highly recurrent enhancer mutations, was among the list. *UBE2V2*, also associated with a highly recurrent indel, is also a dependency gene and moreover, it is an MSI-specific dependency gene in CRC. We further identified *ODGH* as a MSI-specific dependency gene activated by an indel in the in vivo liver metastasis experiment. We have added these results (See Figure 3E, and edits to section titled “MSI-enhancer mutations are recurrent”), and we agree with the reviewers that the inclusion of these data strengthens the manuscript. The data are consistent with a model in which the indels can activate genes that confer an additional fitness advantage following that conferred by the initiating oncogenic driver events. We clarify this model in the Discussion.

With respect to FOX factors, we agree that it is reasonable to hypothesize that MSI CRCs might be more susceptible to FOX knockdown. We looked into this using DepMap data. FOXJ1 was the only FOX factor that shows a significant dependency difference between MSI and MSS (more essential in MSI versus MSS). We are not exactly sure what to make of this result, especially because even though there is a significant difference, the median dependency score for FOXJ1 (at -0.2) is not all that impressive. There could be FOX-specific dependencies that are unique to individual MSI cell lines, and we see some evidence of this. FOXQ1 for example, has a stronger dependency score in LoVo cells than most other cell lines, including many that are MSS. It is also important to note that the motifs enriched among the indels match a variety of FOX factors. Therefore, the silencing of just one might not lead to a dramatic effect on gene expression or cell viability. We further note that, despite often being called pioneer factors, full enhancer activation requires a variety of cofactors, and the knockdown of just one is often insufficient to affect enhancer activity. Combinations are usually required. Additional functional studies that go beyond the scope of this study are clearly required to sort this out. We have edited the text to make this point clearer.

2) More comprehensive demonstration of the statistical associations.In two key places the authors make statements about association (or lack thereof) to chromatin state, but either do not show the data or choose a partial view:• The authors mentioned that enhancer regions overlapped with a large percentage of H3K4me1 ChIP-seq, but the data has not been shown. The authors should add the data supporting for this claim.

We have added aggregate plots of H3K27ac and H3K4me1 ChIP-seq signals at the indel gained enhancers (Figure 1—figure supplement 1A). Given the limited number of peaks, the average signal is not as smooth as one might expect, but clearly shows enrichment of these enhancer histone marks.

• On Figure 2A, the authors claimed that there were no associations between mutation rate and H3K27ac regions in MSI samples. Given that the image on Figure 2A shows the distribution of H3K27ac peaks and mutations specific to chromosome 14, the authors' conclusion would be better supported with a genome-wide association analysis of H3K27ac levels and indels. This should be readily rectified.

We have now included the genome-wide analyses (see Figure 2—figure supplement 1A-B). The results support our initial conclusions.

3) Explanation of the analysis of tumors from mice from different locations.In Figure 6, the authors injected CRISPR-Cas9 induced MMR CRC cells, and control parental lines, into the portal vein of mice to allow for tumor development and downstream epigenomic/genomic analysis. The authors described that CRISPR-Cas9 induced MMR CRC tumors were harvested from liver, while control parental tumors were harvested from peritoneum. However the authors did not justify why tumors at different locations were chosen for the analysis. In addition, the authors did not investigate possible changes on liver tumor development or tumor histology that could indicate changes to oncogenesis induced by increased enhance activity or mutations. A more in depth analysis of these points could provide further support to the authors final hypothesis.

We apologize for the lack of clarity. The MMR-deficient cells grafted to the liver. The MMR-proficient cells grafted to the peritoneum. This may indicate that the MMR phenotype confers a metastatic advantage, although the sample size is too small to confirm this. Despite this difference, we were able to find numerous examples of indel-activating enhancers that were unique to individual liver metastases, supporting the notion that it is the mutation that is activating the enhancer. In other words, to control for site-specific differences, we not only compared the liver mets to the peritoneal tumors, but also the liver tumors to one another. We clarify this in the manuscript. We agree that a comparison between MSS-liver tumors and MSI-liver tumors would be ideal and potentially informative, but the MSS cell lines didn’t form any liver tumors over the course of 3 months in any of 6 mice (of two different genetic backgrounds) that were seeded with these cells.